# A Review of CAR-T Combination Therapies for Treatment of Gynecological Cancers

**DOI:** 10.3390/ijms25126595

**Published:** 2024-06-15

**Authors:** Valentina Olifirenko, Nikolai A. Barlev

**Affiliations:** Department of Biomedical Studies, School of Medicine, Nazarbayev University, Astana 010000, Kazakhstan; valentina.olifirenko@nu.edu.kz

**Keywords:** gynecological cancer, CAR-T, immunotherapy, chemotherapy, p53

## Abstract

CAR-T cell therapy offers a promising way for prolonged cancer remission, specifically in the case of blood cancers. However, its application in the treatment of solid tumors still faces many limitations. This review paper provides a comprehensive overview of the challenges and strategies associated with CAR-T cell therapy for solid tumors, with a focus on gynecological cancer. This study discusses the limitations of CAR-T therapy for solid tumor treatment, such as T cell exhaustion, stromal barrier, and antigen shedding. Additionally, it addresses possible approaches to increase CAR-T efficacy in solid tumors, including combination therapies with checkpoint inhibitors and chemotherapy, as well as the novel approach of combining CAR-T with oncolytic virotherapy. Given the lack of comprehensive research on CAR-T combination therapies for treating gynecological cancers, this review aims to provide insights into the current landscape of combination therapies for solid tumors and highlight the potential of such an approach in gynecology.

## 1. Introduction

Gynecological cancers, including ovarian, cervical, and uterine cancers, remain one of the significant causes of morbidity and mortality among women worldwide. Although there has been major progress in conventional treatment methods, including surgical intervention, chemotherapy, and radiation, the majority of patients diagnosed with advanced or refractory gynecological cancers have a poor prognosis. Development of the resistance to chemotherapy and heterogeneity of gynecological tumors are one of the major challenges in the treatment of such tumors [1,2]. In recent years, there has been growing interest in developing novel immunotherapy approaches to overcome these obstacles in treating gynecological cancers.

CAR-T cell therapy is among the most promising immunotherapeutic strategies reported in recent years. CAR-T is a method of modifying a patient’s own T cells to express chimeric antigen receptors, designed to recognize specific antigens expressed on the surface of tumor cells. Chimeric antigen receptors (CARs) include an antigen-binding domain, T cell receptor (TCR) signaling domains, and various co-stimulatory molecules [3]. Due to the modular nature of cell surface signaling proteins, protein engineering allows for the combination of different extracellular targeting domains and internal signal transduction domains. This feature allows for higher specificity of tumor targeting. CAR-T therapy has already shown promising results in treating blood cancers such as leukemia and lymphoma. Clinical trials of CAR-T cell therapy showed significant remission rates and the possibility for long-term remission. A phase 2 Elara trial of the first FDA-approved CAR-T therapeutic, Kymriah, has shown that out of 94 patients with relapsed/refractory follicular lymphoma, with an overall response rate of 86%, 66% of patients had a complete response [4]. These promising results were further supported by studies in chronic lymphocytic leukemia, multiple myeloma, and large B-cell lymphoma [5,6,7,8].

Despite showing exceptional success in treating hematologic malignancies, CAR-T therapy’s application to solid tumors remains challenging. The limitations of this approach include rapid exhaustion of T cells, poor infiltration into the tumor microenvironment, tumor immune evasion, and rewiring of the tumor microenvironment to promote tumor growth, which is often seen in gynecological tumors. Combination of CAR-T with other therapeutics such as checkpoint inhibitors or chemotherapy is one way to overcome the challenges associated with CAR-T therapy for solid tumors. One innovative approach to enhance the efficacy of CAR-T therapy in solid tumors is the combination with oncolytic viruses [9]. Oncolytic viruses are genetically modified viruses that selectively infect and replicate in tumor cells, leading to their destruction. By combining CAR-T therapy with oncolytic viruses, it is possible to harness the synergistic effect of both and increase the efficacy of treatment.

In this review, we focus on multiple challenges associated with the application of CAR-T in the treatment of solid tumors, including gynecological cancers. We also discuss options for combination therapies with CAR-T, such as checkpoint inhibitors and chemotherapy, as well as the potential use of oncolytic viruses. We explore the rationale behind using this combinatorial approach to overcome the limitations of CAR-T therapy for gynecological cancers.

## 2. Challenges of Using CAR-T Therapy for Solid Tumors

### 2.1. Fast T-Cell Exhaustion

Solid tumors possess several features hindering the function of CAR-T, creating a challenge in the treatment of these cancers. One of such challenges is the fast exhaustion of CAR-T cells within the tumor microenvironment. Due to the immunosuppressive nature of the tumor microenvironment, CAR-T’s cell function is inhibited, limiting its ability to effectively target and destroy tumor cells. This process involves the upregulation of inhibitory receptors such as programmed cell death 1(PD-1), lymphocyte-activation gene 3 (LAG-3), and T cell immunoglobulin and mucin-domain containing-3 (TIM-3) on the surface of T cells [10]. Additionally, the presence of immunosuppressive cells, such as regulatory T cells and myeloid-derived suppressor cells, further contributes to the immunosuppressive environment and the exhaustion of CAR-T cells [11]. Their presence in tumor microenvironment deprives CAR-T cells of essential nutrients and cytokines, leading to their functional impairment and eventual exhaustion. Moreover, the larger tumor area and antigen heterogeneity within solid tumors contribute to the continuous stimulation and faster exhaustion of CAR-T cells. The hypoxic and acidic conditions within solid tumors also negatively impact the functionality of CAR-T cells, contributing to their rapid exhaustion [12]. Numerous studies have also highlighted the role of multiple transcription factors as contributors to CAR-T cell exhaustion within solid tumors; these factors include Thymocyte selection-associated high mobility group box protein (TOX), interferon regulatory factor (IRF4), Nuclear factor of activated T cells (NFATs), and Myeloblastosis transcription factors (MYBs) [13,14,15,16]. In addition, not only long-term resistance but also short-term tumor adaptation to CAR is a problem. A study by Zhai et al. showed the short-term tumor adaptation mechanism involving trogocytosis, during which tumor cells acquire chimeric antigen receptor (CAR) molecules from CAR-T cells. This leads to CAR molecule depletion, dysfunction of CAR-T cells, short-term antigen loss, and antigen masking [17]. All these mechanisms, associated with the presence of antigens, CAR sensitivity, and solid tumor microenvironment, emphasize dynamic processes that impact the effectiveness of CAR-T therapy in solid tumors. Targeting these factors may help to overcome CAR-T challenges in solid tumors and modify the clinical approach to their treatment.

### 2.2. Stromal Barrier

Poor infiltration of CAR-T cells in solid tumors is another major limitation. The stroma of solid tumors, in particular, is a major obstacle that hinders the penetration of CAR-T cells. It is a complex network comprised of various components, including cancer-associated fibroblasts, extracellular matrix proteins, and various signaling molecules, all acting as physical and biochemical barriers that prevent CAR-T cells’ mobilization [18]. Notably, cancer-associated fibroblasts (CAFs) play an essential role in the stromal network in solid tumors [19]. Studies have shown that CAFs produce more collagen types than standard fibroblasts, enhancing the dense and fibrotic nature of the extracellular matrix that creates a physical barrier for the migration of CAR-T cells to the tumor site [20]. Additionally, the stromal network produces various signaling molecules, including cytokines, chemokines, and growth factors. Secretion of Transforming growth factor beta (TGF-β) and interleukin 10 (IL-10) by CAF contributes to the immunosuppressive and pro-tumorigenic milieu, further hindering the infiltration and function of CAR-T cells within the solid tumor [19,21]. Overall, the components of tumor stroma not only create physical obstacles that limit the movement of immune cells but also promote an immunosuppressive environment and tumor evasion.

### 2.3. Antigen-Shedding

Tumor immune evasion represents one of the most substantial factors complicating effective CAR-T cell delivery to cancer patients within the tumor microenvironment. There are several ways through which tumors are able to evade immune response, where one of the most prominent is antigen shedding. Antigen shedding is the process when tumor cells are able to release tumor-associated antigens into the extracellular space, therefore decreasing the number of antigens on the tumor surface available to T cells. For instance, in previous research, a strong correlation was established between the soluble form of signaling lymphocytic activation molecule family (SLAMF7) shed from plasma cells and poor prognosis in multiple myeloma patients [22]. Thus, antigen shedding represents a serious obstacle for CAR-T cell therapy, as it diminishes the recognition and targeting of tumor cells by CAR-T cells. For example, it was demonstrated that glypican-3 (sGPC3) shed in hepatocellular carcinoma (HCC) inhibited the activation and cytotoxicity of CAR-T cells both in vitro and in vivo by acting as dominant negative regulators [23]. This highlights the critical role of antigen shedding in evading CAR-T cell-mediated immune responses. This was supported by Raffaghello et al., who identified that the presence of a soluble form of a Natural killer group 2D (NKG2D) ligand–MHC class I polypeptide-related sequence A (MICA) downregulated surface NKG2D in normal peripheral blood CD8(+) cells, potentially decreasing NK-mediated killing of MICA(+) neoblastoma cells [24]. Notably, the upregulation of proteases by tumor cells contributes to antigen shedding through the cleavage of tumor-associated antigens from the cancer cell surface. For instance, upregulated matrix metalloproteinases are able to proteolytically cleave ligands of NKG2D, such as MICA, UL16 binding protein 2 (ULBP2), and MHC class I polypeptide-related sequence B (MICB), resulting in their shedding and poor recognition by CAR-T cells [25,26].

To overcome limitations due to tumor evasion, new innovative strategies are currently being investigated to enhance the infiltration and function of CAR-T cells in solid tumors.

## 3. Strategies to Enhance CAR-T Therapy in Solid Tumors

Research of multiple challenges of CAR-T therapy for solid tumors has led to the exploration of various approaches aimed at overcoming these challenges and enhancing the efficacy of this promising immunotherapy in solid tumor treatment.

### 3.1. Combination with Checkpoint Inhibitors

Checkpoint inhibitor therapy is one of the promising approaches to be used in combination with CAR-T. It involves preventing tumor cells expressing programmed death-ligand 1 (PD-L1) from interacting with PD-1, which is expressed on the surface of T cells [27]. Given that this approach aims at preventing tumor evasion and stimulating immune response, it can complement the cytotoxic effects of CAR-T cells (Figure 1). Previous preclinical evidence supported the synergistic benefits of combining both therapies. A study by John et al. demonstrated that the administration of anti-PD-1 antibodies together with CAR-T cell therapy resulted in enhanced tumor regression and prolonged survival in a murine solid tumor model [27]. A combinatory approach improved the infiltration of CAR-T cells into the tumor microenvironment and prolonged T cell function due to the immunologically active microenvironment. A strategy utilizing a bicistronic lentiviral vector to induce expression of an anticarbonic anhydrase IX CAR and anti–PD-L1 scFv antibodies in primary human T cells resulted in increased accumulation of anti–PD-L1 antibodies in the tumor microenvironment, blocking PD-1 signaling [28]. These engineered CAR T cells exhibited reduced expression of inhibitory receptors and enhanced effector activity in an orthotopic model of human renal cell carcinoma, suggesting a promising approach for improving CAR T cell efficacy.

Moreover, strategies to block “don’t eat me” signals that are mediated by the CD76/CD86 receptor on the tumor cells could result in unmasking the tumor cells, leading to increased susceptibility to cytotoxic activity of CAR-T cells. “Don’t eat” me signals are often expressed on the surface of tumor cells, acting as a protective mechanism against recognition and phagocytosis by immune cells [30]. Inhibition of interaction between these “don’t eat me” signals and their corresponding receptors, such as the SIRPα receptor on macrophages, can largely increase the susceptibility of tumor cells to immune-mediated cytotoxicity. Previous research suggests that the inhibition of “don’t eat me” would strengthen the CAR-T cell therapy’s efficacy in solid tumors. For example, Dacek et al. demonstrated that CAR-T-secreting antibodies blocking the CD47-SIRPα axis, a prominent “don’t eat me” signaling pathway, were more effective than standard CAR-T therapy [31]. CD47-SIRPα inhibitors effectively promoted the phagocytosis of tumor cells by macrophages, thereby augmenting the overall anti-tumor immune response [31]. This approach not only sensitized tumor cells to macrophage-mediated clearance but also improved the activity of CAR-T cells by reducing the immunosuppressive microenvironment. Furthermore, the combination of CD47 blockade with CAR-T cell therapy has shown promise in preclinical models of solid tumors. By alleviating the inhibitory signals that shield tumor cells from immune surveillance, CAR-T cells are provided with a more favorable milieu for exerting their cytotoxic effects [32]. In a study by Chen et al., the blockade of CD47 in combination with CAR-T cell therapy resulted in enhanced tumor regression and prolonged survival in a murine model of solid tumors [33]. The combination approach not only facilitated the infiltration of CAR-T cells into the tumor microenvironment but also promoted a pro-inflammatory response, further amplifying the anti-tumor efficacy.

Expression of PD-1 and PD-L1 has been found in more than half of tumor stroma samples and the majority of cervical cancer tissue samples but is seldom seen in normal cervical tissues [34,35]. Thus, PD-1/PD-L1 can serve as promising drug targets for more effective treatment of gynecological cancers. The clinical trials using PD-1 blockers such as pembrolizumab and nivolumab as a treatment for cervical cancers demonstrated a remarkable median overall survival and prolonged progression-free survival. As a result, pembrolizumab has been approved by the FDA for the treatment of cervical cancer. However, studies assessing the application of pembrolizumab and nivolumab for ovarian cancers have shown inconsistent results, with some data indicating no effect, while others highlight an improved overall response rate [36,37]. Further research is needed to fully understand the effectiveness of PD-1 blockers in treating cervical and ovarian cancers, as well as application in other gynecological tumors. Nevertheless, high immunogenicity of gynecological tumors suggests potential use of checkpoint inhibitors in combination with CAR-T to improve its effectiveness even though more research is required to understand the underlying molecular mechanisms of PD-1 inhibition in gynecological tumors.

Overall, immune checkpoint inhibitors and blockade of “don’t eat me” signals represent an exciting strategy to support the application of CAR-T cell therapy in solid tumors. As this approach unleashes immune recognition and clearance of tumor cells, it is theoretically likely to work in synergy with the CAR-T cell-mediated cytotoxicity, which could turn down multiple resistance strategies used by solid tumor microenvironments.

### 3.2. Common Immune-Related Adverse Effects of Checkpoint Inhibitors

Despite promising results of checkpoint inhibitors in the treatment of various types of cancer, they can also lead to immune system overactivation, which can manifest even stronger due to their synergistic effect with CAR-T. The most common immune-related adverse effects due to checkpoint inhibitors include colitis, neurotoxicity, hepatitis, inflammatory dermatitis, lymphopenia, and other inflammatory diseases [38]. Simultaneous use with CAR-T can potentially exacerbate these side effects, as well as lead to cytokine release syndrome—a systemic inflammatory response [39]. The severity of these side effects can range from mild to life-threatening; therefore, each case must be approached with care. For management of mild adverse effects such as nausea or mild skin reactions, monitoring by a physician and over-the-counter medications might be sufficient [40]. Moderate to severe side effects require additional interventions and occasionally even complete discontinuation of therapy. System inflammation syndromes like severe colitis, hepatitis, or osteoporosis are often mediated by immunomodulators such as corticosteroids, immunosuppressive antibodies (e.g., tocilizumab), and other inhibitors (e.g., cyclosporine, azathioprine, methotrexate) [40,41]. Tocilizumab, a monoclonal antibody that targets the interleukin-6 receptor, was also shown to be effective in the case of severe cytokine release syndrome that can occur due to a combination of immune therapies [39]. Overall, while the combination of checkpoint inhibitors and CAR-T cell therapy holds great potential, it requires careful monitoring by physicians to alleviate potential immune-related side effects.

### 3.3. Integration with Chemotherapy

Chemotherapy remains an essential approach for the treatment of gynecological cancers, particularly for advanced-stage cases where limited treatment options are available. In this respect, it is important to mention that the status of the human tumor suppressor p53 seems to be critical for the efficacy of chemotherapy [42,43].

Dysfunction of p53 is observed in many gynecological malignancies. This may be due to the direct effect of mutations in the tumor protein 53 (TP53) gene or can be mediated by the dysregulation of enzymes that control the stability and transcriptional activity of the p53 protein [44,45]. In cervical cancer, p53 is degraded at the protein level due to its interaction with the ternary complex between human papillomavirus (HPV) E6, E6-associated protein (E6AP), and E3 ubiquitin ligase UBE3A [46]. In endometrial cancer, overexpression of mutant p53 in immunohistochemistry is a significant prognostic sign. A discrepancy between p53 overexpression and mutations in the TP53 gene is observed in endometrial cancer, indicating that the accumulation of the p53 protein may be explained not only by gene mutations in p53 but also by dysregulation of factors that control the p53 stability, e.g., p53-specific E3 ligase, Mouse double minute 2 homolog (MDM2). Accordingly, the restoration of the wild-type status of p53 using small molecules in several types of gynecological cancer led to increased sensitivity of tumor cells to chemotherapy [47,48].

However, its effectiveness is hindered by the lack of target selectivity, as it can impact both cancerous and healthy cells. This results in adverse effects on other organ systems, which significantly impairs the quality of patients’ life and has a high chance of resistance development [49]. In recent years, there has been growing interest in combining chemotherapy with CAR-T cell therapy as a potential treatment approach for gynecological cancers. The rationale behind this approach is based on the potential synergistic effects of chemotherapy in priming the tumor microenvironment to be more receptive to CAR-T cell infiltration and function. Administration of both therapies simultaneously will enable us to apply lower doses of chemotherapy, therefore minimizing its toxic side effects. In this case, the necessary synergism will be achieved by changing the tumor environment with the chemotherapy effect and specifically targeting cells with CAR-T cells (Figure 2).

By inducing immunogenic cell death and releasing tumor-associated antigens (TAAs), chemotherapy augments antigen-presenting cell (APC) antigen presentation, which in turn activates and enhances effector T cell activity, including that of CAR-T. Multiple studies have shown that chemotherapy-induced release of tumor antigens and damage-associated molecular patterns promotes the recruitment and activation of tumor-infiltrating lymphocytes, which may synergize the cytotoxic activity of CAR-T cells within the tumor microenvironment [50,51,52]. Chemotherapy has also been found to have immunomodulatory effects where traditional immunosuppressive cells including regulatory T and myeloid-derived suppressor cells are depleted, which may counteract inhibitory signals in the tumor microenvironment. For example, traditional chemotherapy agents, such as cyclophosphamide and fludarabine, are known to reduce the population of immunosuppressive cells in solid tumors [53,54]. This may relieve the suppressive milieu and enable increased CAR-T cell penetration and function within tumors. Moreover, chemotherapy is cytotoxic and capable of causing tumor debulking, creating physical space in the tumor microenvironment, and altering the tissue structure, which may enable improved CAR-T cell penetration and recruitment. The study by Alvarez et al. demonstrated a lower number of CAFs and less fibrillar collagen in tissues of mice treated with nab-paclitaxel, which could enhance CAR-T cell infiltration and distribution within the tumor microenvironment [55].

One notable disadvantage, however, is the potential for increased systemic toxicity when combining chemotherapy with CAR-T cell therapy [52]. Chemotherapy is often associated with severe toxicity toward healthy tissues and organs, manifesting in hematologic and gastrointestinal symptoms and immunosuppression. Combined use of chemotherapy with CAR-T can increase the overall toxicity for organs undermining the overall tolerance to the treatment. Additionally, a common side effect of many chemotherapeutics is lymphodepletion, which affects both endogenous T cells and infused CAR-T, thereby reducing their efficacy [52].

These limitations and potential challenges suggest that while the standard improvements in immunotherapy and integration with chemotherapy are beneficial, they require an individualized approach. Standard immunotherapy techniques and checkpoint inhibitors may cause immune-related and systemic toxicity, as well as various levels of chemotherapy drug resistance. However, studies on the novel oncolytic viral therapy report less overall systemic side effects, most of which can be managed with standard hospital care [56]. Therefore it is possible to suggest that the use of oncolytic viral therapy in conjunction with CAR-T can potentially be a more effective option for the treatment of solid tumors.

## 4. Novel Oncolytic Viral Therapy Combined with CAR-T

### 4.1. Prospects of Viral Therapy Combination

In recent years, novel oncolytic viral therapy combined with CAR-T therapy has gained attention as a more effective treatment for cancer. Research studies have highlighted the capacity of oncolytic viruses to improve the efficiency of CAR-T cell therapy through various mechanisms, such as tumor priming, immune modulation, and enhanced tumor targeting [57,58,59] (Figure 3).

Oncolytic viruses are able to infect and selectively replicate within tumor cells, causing tumor cell lysis and the release of tumor-specific antigens. This process promotes immunogenic killing of these cells, thereby aiding the immune system in identifying the antigens of the tumor cells, as well as releasing danger signals and chemokines that attract immune effector cells, including CAR-T cells, to the tumor site. Previous examples, such as oncolytic H-1 parvovirus, show safe applications and signs of immunogenic activity in the treatment of solid tumors [60,61]. The immunostimulatory milieu generated by oncolytic viruses can serve as an ideal “inflammatory” environment for subsequent CAR-T cell infiltration and activation within the tumor microenvironment. Furthermore, oncolytic viruses are known to target and remove immunosuppressive cell populations, further making the environment more immunogenic. For instance, several oncolytic viruses have been engineered to express immunomodulatory proteins that can directly target regulatory T cells and myeloid-derived suppressor cells. The study by Wing et al. investigated the combination of chimeric antigen receptor T (CART) cells targeting the folate receptor alpha (FR-α) with an oncolytic adenovirus armed with an EGFR-targeting, bispecific T cell engager (OAd-BiTE) in solid tumors [62]. Upon infecting tumor cells, the oncolytic virus can secrete EGFR-targeting BiTE, which, in turn, promotes specific and enhanced T cell activation. This demonstrates that oncolytic viruses work to remodel the tumor microenvironment in synergy with CAR-T cell therapy. Due to the localized effect of intratumoral administration of oncolytic viruses, CAR-T cells are more likely to penetrate and destroy tumors. For instance, it was shown that combining mesothelin-redirected chimeric antigen receptor T cells (meso-CAR T cells) with an oncolytic adenovirus expressing TNF-α and IL-2 (OAd-TNFa-IL2) significantly enhanced antitumor efficacy in pancreatic ductal adenocarcinoma (PDA) models. The combination of two approaches helped in overcoming the immunosuppressive tumor microenvironment and improving T cell function, suggesting a promising approach for the treatment of PDA [57]. Notably, oncolytic viruses remodel the tumor stroma by enhancing vascular permeability and increasing nutrient supply, which is essential for establishing CAR-T cells’ function and persistence within the tumor [58]. The preclinical studies of oncolytic viruses in combination with CAR-T cell therapy have already shown promising results, including longer CAR-T persistence, improved targeting, and enhanced cytotoxic effects [59,63,64,65].

Due to its unique property to infect only cancerous cells and induce a local immune response against the tumor, an oncolytic virus is an attractive option for combined therapy. In recent years, considerable progress has been made in the development and application of oncolytic viral therapy for gynecologic malignancies. For instance, Benencia et al. demonstrated that the use of oncolytic herpes simplex virus type 1 (HSV-1) in patients with recurrent gynecologic cancer resulted in the regression of the tumor and prolonged relapse-free survival [66]. Another study by Kim et al. on murine models revealed that oncolytic therapy based on adenovirus with deleted E1B 19kDa and E1B 55kDa genes showed significant tumor growth suppression [67].

Consequently, the combination of CAR-T cell therapy and oncolytic viruses might have several synergistic benefits in treating gynecological cancers. Firstly, oncolytic viruses can be engineered to specifically infect and replicate within target tumor cells and release tumor antigens upon lysis, boosting the activity of CAR-T cells to recognize tumor antigens. Secondly, the immunostimulatory potential of oncolytic viruses may make the tumor microenvironment less suppressive for CAR-T cells and prevent their exhaustion. Finally, the overall systemic immune responses triggered by oncolytic viruses might help enhance the cytotoxic activity of CAR-T cells.

In conclusion, the synergistic action of CAR-T cells and oncolytic viruses offers immense opportunities for the management of gynecological cancers in theory. However, due to a lack of research in the field, more experiments are still needed to determine the optimal and most effective way of using CAR-T cell therapy in conjunction with oncolytic viruses.

### 4.2. Challenges with Viral Therapy Combination

Even though a combination of oncolytic viral therapy with CAR-T cells offers a hopeful approach for overcoming the challenges associated with solid tumors, it is essential to acknowledge and resolve the specific limitations of viral therapy when used in combination with other therapeutic options.

The risk of off-target effects possible for any methodology and systemic viral therapy toxicity can be compared to conventional chemotherapy. Furthermore, oncolytic viruses and CARs can both cause a systemic immune response that may result in adverse effects such as crush syndrome and neurotoxicity. Simultaneous treatment with oncolytic viruses and CAR-T cells may exacerbate these toxicities, which will be a significant issue in controlling and minimizing the side effects [9].

Another drawback of a combination approach could be that interaction between oncolytic viruses and CAR-T cells within the tumor microenvironment may lead to competition for resources and space, thereby impacting the persistence and functionality of CAR-T cells. The proinflammatory setting induced by oncolytic viruses might also activate regulatory pathways, reducing the CAR-T cells’ proinflammatory activity and undermining their antitumor efficacy. The study by Evgin et al. found that oncolytic virus infection (VSVmIFNβ) led to the attrition of murine EGFRvIII CAR T cells in a B16EGFRvIII murine model, despite inducing proinflammatory chemokines secretion. The underlying therapeutic interference presents an unexpected mechanism that necessitates further study to enhance the synergistic effect of chimeric antigen receptor T cells in solid tumors combined with oncolytic viruses [68].

Despite these limitations, there are several potential strategies to counteract these disadvantages regarding toxicity for the combinatory use of oncolytic viruses and CAR-T cell therapy. One potential method to overcome the off-target side effects of oncolytic viruses and enhance the safety of this therapy when combined with CAR-T cells involves achieving better targeted delivery of the viral vector. By enhancing tissue-specific targeting and minimizing the systemic dispersion of viruses through engineered serotypes or intratumor delivery routes, the off-target toxicity of viral vectors can be reduced. Severe or prolonged adverse events due to immune overactivation can be managed by the utilization of immunomodulators such as tocilizumab or corticosteroid therapy. Additionally, Zheng et al. suggest that melatonin can be used as an effective immunomodulator for mitigation of CAR-T-induced cytokine release syndrome without compromising the anti-tumor activity of CAR-T [69]. Therefore, it can be suggested for use in combination therapy as well.

## 5. Conclusions

In conclusion, the treatment of gynecological tumors using CAR-T in combination with other immune therapies, including checkpoint inhibitors, chemotherapy, and oncolytic viruses, represents a promising measure to address the existing challenges. Despite the fact that there has been limited research on the application of combination therapies for gynecological cancers, existing research in other solid cancer types highlights that this approach can be highly effective. Combination of CAR-T with oncolytic viruses stands out as a promising strategy due to the high possibility of synergistic effects between the two treatments. Oncolytic viruses have shown promise in targeting and destroying cancer cells, and when combined with CAR-T therapy, they could potentially enhance the overall anti-tumor immune response. However, the risks associated with systemic toxicity and immune overactivation are yet to be assessed. Overall, while more research is needed, the combination of CAR-T therapy with oncolytic viruses, checkpoint inhibitors, or chemotherapy holds great promise for improving outcomes in the treatment of gynecological tumors.

## Figures and Tables

**Figure 1 ijms-25-06595-f001:**
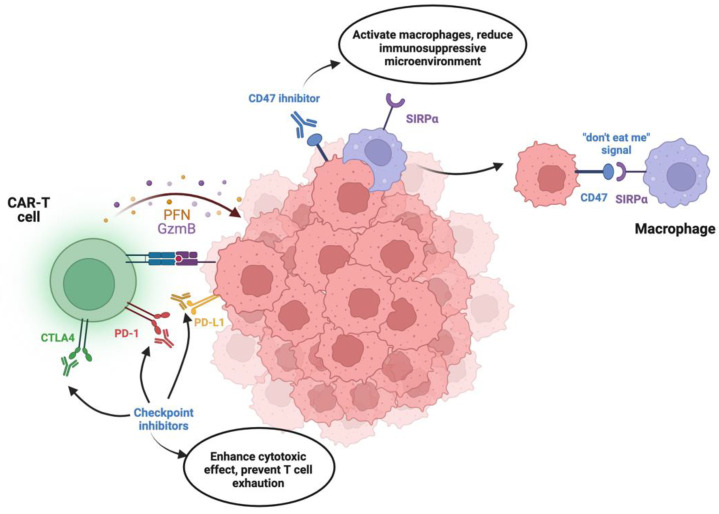
Combination therapy of checkpoint inhibitors and CAR-T [29]. Checkpoint proteins such as PD-1 and cytotoxic T-lymphocyte-associated protein 4 (CTLA4) work as regulators suppressing the immune response upon binding to their ligands (PD-L1, CD80, and CD86). Inhibition of checkpoint proteins removes this immunosuppressive blockage for T cells and, in turn, allows CAR-T to have a prolonged and more stable cytotoxic effect through perforin (PFN) and granzyme B (GzmB). Additionally, the presence of “don’t eat me” signals such as CD47 on the surface of tumor cells inhibits macrophages immune function upon binding to signal regulatory protein α (SIRPα). Inhibitors of CD47 prevent immune evasion from macrophages, thus making the tumor microenvironment more suitable for CAR-T cell function.

**Figure 2 ijms-25-06595-f002:**
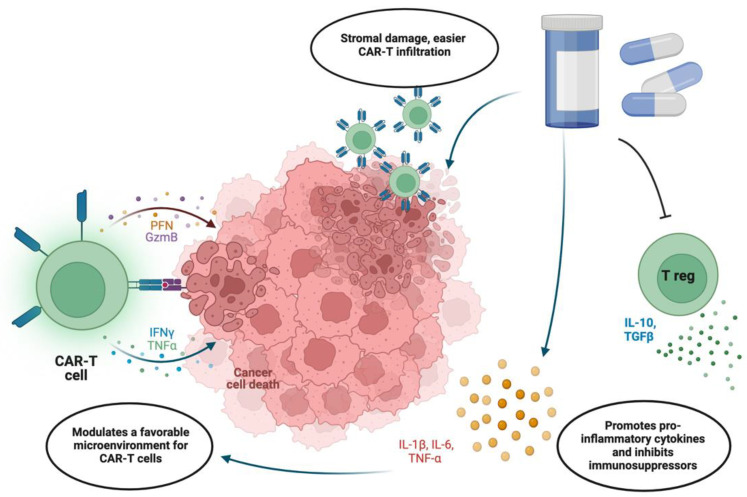
Combination of chemotherapy and CAR-T [29]. Chemotherapeutic drugs are able to reduce tumor burden as well as damage tumor stroma, allowing CAR-T cells to better infiltrate solid tumors. Additionally, many drugs have been shown to reduce the number of T regulatory cells that promote an immunosuppressive environment through IL-10 and TGFβ. Moreover, chemotherapy promotes the release of immunogenic chemokines such as IL-1β, IL-6, and TNFα, creating a more favorable microenvironment for CAR-T function.

**Figure 3 ijms-25-06595-f003:**
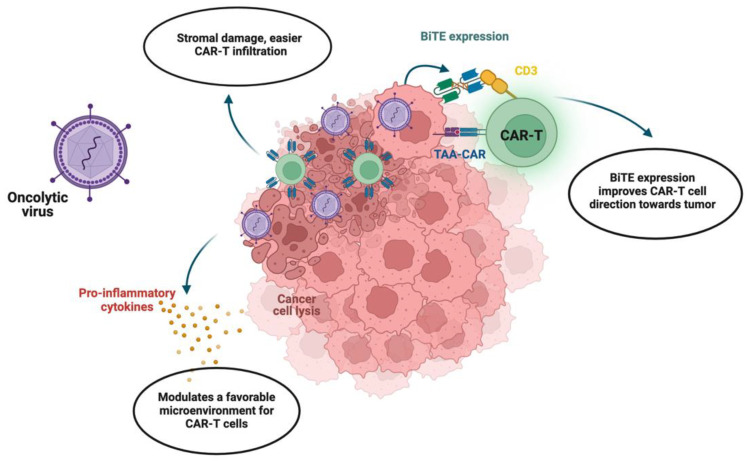
Combination of oncolytic viruses and CAR-T therapy [29]. Infection of tumor cells with oncolytic viruses leads to lysis of tumor cells, resulting in damaged stroma, release of cancer antigens, and promotion of immunogenic microenvironment through pro-inflammatory cytokines. This allows for better infiltration and prolonged function of CAR-T cells. Additionally, oncolytic viruses can be engineered to modify cancer cells and promote bispecific T cell engager (BiTE) expression, which enables targeting of CAR-T cells compared to standard TAAs. This enhances CAR-T cell specificity and function.

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
