# Peer review of "A Review of CAR-T Combination Therapies for Treatment of Gynecological Cancers"

_ijms, 2024, doi:10.3390/ijms25126595_

Round 1

Reviewer 1 Report

Comments and Suggestions for Authors

Reviewer statement:

CAR-T Combination Therapy for Treatment of Gynecological Cancers

Gynecological cancers, including ovarian, cervical, and uterine cancers, remain one of the significant causes of morbidity and mortality among women worldwide. Improvement of management of gynecological cancers can improve prognosis, reduce morbidity and enhance quality of life. The introduction of novel immunotherapy approaches is expected to improve treatment and prognosis, although there is still a lot to know about the new approaches. Chimeric antigen receptors (CAR)-T cell therapy is considered one of  the most promising immunotherapeutic strategies in recent years. The authors conducted a review focusing on the challenges associated with the application CAR-T in the treatment of solid tumors, including gynecological cancers, discuss the options for combination therapies with CAR-T and explore the rationale behind using combinatorial approach to overcome the limitations of CAR-T therapy for gynecological cancers,  which is interesting, important and clinical relevant. This study can add valuable information, adding data to the available knowledge  and research on CAR-T therapy.

Title: The title reflects the study reported and the type of study conducted, which is excellent.

1.       The authors could add a review to the title to make it clear that they are reporting a review.

Overall: The paper is well written and attractive to read. The authors should be complimented for doing this.

2.       The authors use a lot of abbreviation which are not always written out completely the first time in the article ( examples line 74, 84). Please check the entire article on abbreviations. The use of a list with the use abbreviation might be useful to the reader.

The sections 1-4 and conclusion were very attractive to read form a reader point of view, explaining the background, reason for conducting this study, the rationale of combination therapy,the struggles with CAR-t therapy and the potential of CAR-T therapy.

It was and excellent review. I want to complement the authors for this achievement.

Figures

Tables : no comment.

Author Response

Thank you for the review of our article on CAR-T Combination Therapy for Treatment of Gynecological Cancers. We appreciate your positive feedback and are glad to hear that you found the review to be well-written and engaging.

According to your suggestion we have changed the title to  "A Review of CAR-T Combination Therapies for Treatment of Gynecological Cancers". Additionally, we made sure that all names are written fully the first time they appear in the text with abbreviations in brackets for further use.

Thank you for taking the time to review our work and for your valuable comments. We believe  that these revisions have enhanced the overall quality of the article.

Reviewer 2 Report

Comments and Suggestions for Authors

A good summary of CAR-T cell combinational therapy approaches possible in the future. Also good mentioning, even thought short of the the potential of overactivation of immune responses and unfavorable side effects of autoimmunity.

Few things to check: 

Generally check the use of abbrivations and lacking full name of such e.g.: PD-1 you already mention at 74 but full name is given on 152. no full name given of LAG-3, Tim-3, TOX....

18-21: These two sentences are repetitive and in the second you write "....solid tumors in gynecological cancers." Better write "...solid tumors in gynecology" or such, as gynecological cancers are anyhow solid tumors.

55-56: "One innovative..." please add reference on such a trial or experiment of combining therapies

95: Change the sentence beginning with "Stroma" to "The stroma of solid tumors,..., is a major obstacle..."

98: "all of act.." - all acting

108: not "...as well as..." - "but also..."

144: CD80 ... CD86, and missing?

193: Revise sentence "Thus it is potential..." difficult to understand what you want to say

203: "....suggests potential use"

223: sentence seems to be missing in part: "...essential approach for the treatment of ???, particularly"

247: repetitive use of apply: "Application of both therapies simultaneously will allow applying..." potentially write "Administration of ....

251 -252: TAA: full name missing (in figure 3), APC too

262: "...and allow increased CAR-T cell penetration? Assume the penetration of CAR-T cells is there but insufficient generally

273: "..overall toxicity for the organism undermining" or "...for organs..."

292: "Research studies..." Ref missing which research studies

300: "Previous examples, such as oncolytic H-1 parvovirus, show (not shows)..." safety??? I assume "safe application"

305 "further making the environment more immunogenic."

312-314: "This demonstrates that oncolytic viruses work to remodel the tumor microenvironment in synergy with CAR-T cell therapy." remove second work at end of 313.

316-321: Very long sentence - for better reading turn it into two sentences

322: "..essential for establishing CAR-T cell function and persistence.."

331: Better write "...and prolonged relapse-free survival"

334: delete "...in murine models" as you start the sentence with "...on murine models"

345: due instead of sue

353: "The risk of off-target effects.."

359: "Another disadvantage of ????

386 - 390: Same sentence written twice starting with "Oncolytic viruses..."

In chapter 4.2 a short example for the mentioned immunomodulators given to counter adverse effects would be a good addition. 

The overactivation of immune responses as mentioned at the bottom of 3.1 could be added as a short sub-chapter on it's own before the discussion on which effects are expected and what needs to be specifically countered when such strong side effects are detected. As this will be most likely a difficult challenge in application of CAR-T cells in combination with other immune-modulatory components.

Good summary, some parts are missing either words or phrasing but in general it is a good outlook for therapy applications in solid cancers with CAR-T cells.

Author Response

Thank you for your detailed feedback on the article. We have carefully reviewed your comments and made the necessary corrections according to your suggestions. Here's a summary of the changes that have been implemented:

1. Abbreviations: We have ensured that all names are written fully the first time they appear in the text with abbreviations in brackets for further use.

2. Sentence clarity and mistakes correction: Please find line numbers in original text which you suggested to correct and respective numbers of current modified text.

18 = 18

55 = 55

95 = 99

98 = 103

108 = 113

144 = 151

193 = 201

203 = 211

223 = 249

247 = 274

262 = 291

273 = 302

292 = 330

300 = 335

305 = 341

312 = 348

316 = 351

322 =  358

331 = 369

334 = 369

345 = 382

353 = 390

359 = 396

386 = 428

3. We have incorporated a short example of immunomodulators in chapter 4.2 as per your suggestion (414-419). 

4. We also included a sub-chapter addressing the immune over activation related adverse effects (220-240). This was written about checkpoint inhibitors therapy because it is the main therapy that can cause severe adverse effect due to immune reaction specifically. Chemotherapy and viral therapies mainly have problems of drug toxicity and off-target effects, which is mentioned in their respective chapters. Some immune suppressors mentioned in 3.2. can be used for viral combination therapy which is also referenced in 4.2 (414).

We appreciate your valuable input, and we believe that these revisions have enhanced the overall quality of the article. Thank you for taking the time to review our work.